# Chip-scale reconfigurable carbon nanotube physical unclonable functions

Yang Liu[1,2,5], Jingfang Pei[1,5], Yingyi Wen[1], Lekai Song[1], Songwei Liu[1], Pengyu Liu [1], Wenyu Cui [3], Zihan Liang[4], Teng Ma [3], Xiaolong Chen [4] & Guohua Hu [1] ✉

With the rapid advancement of edge intelligence, ensuring the security of edge devices and protecting their communication has become critical. Physical unclonable functions, known as *hardware fingerprints*, are an emerging hardware security solution enabled with the physical variations inherent in the hardware systems. To facilitate a widespread edge deployment, here we present chip-scale reconfigurable physical unclonable functions built with carbon nanotube charge-trapping transistors, where the charge-trapping memory and physical variations of the transistors are harnessed to render over $10^{13}$ reconfigurable states and the demonstrated ideal physical unclonability. Arising from this, the physical unclonable functions prove robust resilience against advanced machine learning and artificial intelligence attacking (limiting success to ~50–60%) as well as brute force cracking (requesting an estimated $10^{16}$ years to crack). This performance, along with their scalability and low-power operation as well as cryogenic temperature robustness, position the physical unclonable functions a promising hardware security solution for edge intelligence. As a practical demonstration, we model self-driving vehicular network in Central Hong Kong and prove secure vehicle communication using the physical unclonable functions.

As edge intelligence becomes increasingly integral to applications such as the Internet of Things (IoT) and autonomous systems, securing the distributed edge devices has emerged critical[1]. Concerns arise for authenticating the identities of these edge devices, combating the counterfeits, and establishing trust protocols for their communication[1]. Taking self-driving as an example, self-driving requires authentication of the vehicles, alongside secure generation and transmission of sensitive driving - both among the vehicles and between the vehicles and cloud servers. Implementing robust authentication mechanisms to prevent unauthorized access, and adopting secure communication protocols to safeguard the driving data against theft, manipulation, or misuse via reverse engineering,

hacking, and cyberattacks are essential[2]. In manufacturing of self-driving vehicles, consequently, hardware-based security measures are often integrated for reliable vehicle authentication and secure communication[3].

Among the many hardware security measures, physical unclonable functions (PUFs) have emerged as a promising solution[4]. PUFs, known as *hardware fingerprints*, exploit the physical variation inherent in the hardware systems introduced during manufacturing to produce unpredictable responses when challenged[4]. Owing to the entropic nature of the physical variation, the responses are theoretically unique and unclonable, specific to each hardware instance[4,5]. These responses can thus be repeatedly produced to create physical unclonable

[1]Department of Electronic Engineering, The Chinese University of Hong Kong, Shatin, New Territories, Hong Kong SAR, China. [2]Shun Hing Institute of Advanced Engineering, The Chinese University of Hong Kong, Shatin, New Territories, Hong Kong SAR, China. [3]Department of Applied Physics, Hong Kong Polytechnic University, Hung Hom, Kowloon, Hong Kong SAR, China. [4]Department of Electrical and Electronic Engineering, Southern University of Science and Technology, Shenzhen, China. [5]These authors contributed equally: Yang Liu, Jingfang Pei. ✉e-mail: ghhu@ee.cuhk.edu.hk

primitives[4,5]. Current PUF development predominantly relies on integrated silicon circuits, such as those based on arbiter and ring oscillator architectures, realized by the trivial physical variation in the silicon electronics[6]. However, the silicon electronics often suffer from limited entropy, and may thus require substantial hardware and energy overhead for the design and operation of PUFs[7]. To address the limitations and, importantly, to facilitate widespread edge deployment of PUFs in edge intelligence, scalable PUF solutions are being sought, particularly those exploiting solution-processed materials and devices capable of exhibiting high-level entropy[8]. To align with the constrained hardware budget of edge scenarios, reconfigurable PUFs are being explored to enhance the security. A PUF is reconfigurable if it can be updated post-manufacturing through physical mechanisms and operations[8,9].

Carbon nanotubes with high carrier concentration and mobility as well as specific surface area allow dynamic engineering of their electronic structures[10,11]. Herein, in this context, we present chip-scale PUFs developed with solution-processed carbon nanotube charge-trapping transistors, where the PUFs by exploiting the charge-trapping memory and physical variation of the transistors achieve over $10^{13}$ reconfigurable states for generating physical unclonable primitives. Through rigorous evaluation, we prove the ideal physical unclonability of the PUFs, evidenced by their ideal randomness, uniqueness, and irrelevance from one another and in reconfiguration operations. The PUFs with the physical unclonability and reconfigurability demonstrate robust resilience against advanced machine learning and artificial intelligence attacking as well as brute force cracking, manifesting their potential to enable hardware security in edge intelligence. As a demonstration, we model self-driving vehicular network in Central Hong Kong, with PUF-based key exchange protocols embedded to secure the vehicle communication.

## Results

### Chip-scale reconfigurable PUFs

We develop PUFs with carbon nanotube charge-trapping transistors, following the fabrication method in our previous report[12]. Briefly, semiconducting-phase single-walled carbon nanotubes are sorted out in solution with PCz polymer, and deposited to fabricate the transistors via photolithography (Fig. 1a–d). Arising from charge trapping and the dynamics (Supplementary Fig. 1), the transistors achieve reconfigurable non-volatile memory with over 32 easily reconfigurable states (Fig. 1e, f), and the memory is robust at low and even cryogenic temperatures (Supplementary Fig. 2). The transistor fabrication is wafer scalable, with a yield of >98%, and importantly, sampling tests across the wafer prove a high uniformity. For example, the memory window of 300 randomly sampled transistors (Fig. 1e) is 11.40 ± 0.62 V, giving a variation of only 5.4% (Fig. 1g), though the correspondingly extracted mobility and subthreshold are a bit more variant (Fig. 1h, i). These uniform yet variant performance metrics lay the foundation for realizing physical unclonability towards PUF development. The variation across the transistors arises from the random distribution of carbon nanotubes in transistor fabrication. In addition, we demonstrate that the transistors allow cycle-to-cycle stable resetting and reconfiguration operations (Supplementary Fig. 3).

The wafer-scale transistors with the exhibited uniformity and yet trivial physical variation as well as the stable memory reconfiguration suggest the potential in developing PUF. Here we prototype PUF chips as shown in Fig. 2a, with nine transistors interconnected via one shared common drain to create a PUF. See Supplementary Fig. 4 for the optical images of the PUFs. Upon operation, the states of the individual transistors are first reset and then configured as required via the gates (Supplementary Fig. 5). After the configuration, a voltage pulse is applied to the common drain as the challenge, allowing the sources to output parallel current pulses as the response (Fig. 2a). The response carries the variation of the transistors for physical unclonable primitive

generation. To generate the primitives, the response in analog is binarized via an ADC testing board, and each of the response pulses produces 12 binary digits. This gives primitives in 9 × 12 bitmaps, i.e., in a 108-bit length. See Supplementary Fig. 6 for the design and operation of the ADC testing board. Note the 12-bit length analog-digital conversion is defined by the ADC used in our work, and other bit length digits can be generated using other ADC models. Given the reconfigurability of the transistors (Fig. 1f), the PUFs can be easily configured to over $32^9$ states for primitive generation, far outperforming other studies (Supplementary Table 1). As an example, here we show the first PUF, denoted as PUF$_1$, can be configured to different states from the initial state for primitive generation and, importantly, after the configurations, it can be reset to the initial state (Fig. 2b, c). The initial state means all the individual transistors are in the initial high conductance states. The PUFs starting from a random state can still be configured and reset (Fig. 2d).

A practical operation of the PUFs demands stable configuration and primitive generation. This in turn requires the transistors to withstand constant configuration and stable challenge-response generation, as proved in Fig. 1j and k. See also Supplementary Fig. 7 for the highly stable challenge-response generation from the transistors across 100,000 sampling times in the initial and other states. This stable transistor operation can give rise to a limited bit error rate (BER) in PUF primitive generation. Indeed, we demonstrate that the BER in practical operation of our PUFs is <1% (Fig. 2e, Supplementary Fig. 8). Given this, the real bit length of our PUF primitives is 106-bit (i.e., 108 × 99%). Here we note the error bits generated may be further corrected by hardware (e.g., using other high precision ADCs) and algorithm designs (e.g., temporal majority voter, median value by multiple measurements, and error-correcting code; Supplementary Fig. 9)[13]. In addition, the PUFs prove a long lifetime, as demonstrated in Supplementary Fig. 10 where we show the transistors of a typical PUF are tested up to 120 days and demonstrate stable operation throughout the period. The operational stability may, however, extend beyond this.

### Physical unclonability

A practical operation of the PUFs demands ideal physical unclonability. Taking PUF$_1$ as the example, here we examine the entropy of the primitives generated by PUF$_1$ at the initial state (Fig. 3a). Entropy describes the distribution of the 1 s and 0 s bits in the primitives, and can be quantified by $E = -\left[p\log_2 p + (1-p)\log_2(1-p)\right]$, where $p$ is the ratio of the 1 s bits[9]. An entropy of 1 means a random distribution of the primitive bits, i.e., the ideal randomness of the primitives[13]. As demonstrated, the primitives all present an entropy of ~1, proving an ideal randomness. We then analyze the normalized Hamming distance (n-HD). n-HD describes the differences between any two random primitives, and can be quantified by $d_H(x,y) = \frac{1}{n}\sum_{i=1}^{n}1(x_i \neq y_i)$, where $x_i$ and $y_i$ are the corresponding bits in the two primitives[14]. An n-HD of 0.5 represents a maximum uniqueness across the primitives[9]. As demonstrated, the primitives all present an n-HD of ~0.5, proving an ideal uniqueness. See Supplementary Table 2 for the detailed n-HD values (and also other physical unclonability metric values) as estimated. To thoroughly investigate the physical unclonability, we further study the correlation coefficients (CC). CC, denoted by $\rho_{x,y}$, measures the linear relationship between any two random primitives $x$ and $y$, and can be quantified by

$$\rho_{x,y} = \frac{cov(x,y)}{\sigma_x \sigma_y} = \frac{\frac{1}{n}\sum_{i=1}^{n}\left(x_i - \frac{1}{n}\sum_{i=1}^{n}x_i\right)\left(Y_i - \frac{1}{n}\sum_{i=1}^{n}y_i\right)}{\sqrt{\frac{1}{n}\sum_{i=1}^{n}\left(x_i - \frac{1}{n}\sum_{i=1}^{n}x_i\right)^2}\sqrt{\frac{1}{n}\sum_{i=1}^{n}\left(y_i - \frac{1}{n}\sum_{i=1}^{n}y_i\right)^2}},$$

$$(1)$$

where $cov(x,y)$ is the covariance between the two primitives $x$ and $y$, and $\sigma_x$ and $\sigma_y$ are the respective standard deviations[15]. A CC of

0 suggests no linear correlation, i.e., the primitives are irrelevant to one another[9]. As demonstrated, the primitives all present a CC of ~0, proving an ideal irrelevance.

As demonstrated, the entropy, n-HD, and CC metrics all prove an ideal physical unclonability of the PUFs at the initial state. On the other hand, as discussed, to apply the PUFs in edge intelligence, a reconfigurability is expected to adapt to the edge applications and enhance the hardware security. Here we study the physical unclonability of the PUFs at five other randomly configured states (Supplementary Fig. 11) and the final state (Fig. 3b). The final state means all the individual transistors are configured in the final low conductance states. As shown, the entropy, n-HD, and CC all prove almost ideal metrics, i.e., entropy~1, n-HD ~ 0.5, and CC ~ 0. This demonstrates the robust, ideal physical unclonability with reconfigurability of the PUFs. A practical application of the PUFs in edge intelligence would also demand PUF-to-PUF unclonability to ensure the hardware security. Here we analyze

the n-HD across the PUF chip as prototyped (Fig. 3c). As demonstrated, the PUF-to-PUF n-HD matrix is all ~0.5, proving an ideal uniqueness of the PUFs across the PUF chip. Note that the 40 PUFs are all configured at the initial state in this analysis.

Having demonstrated the physical unclonability with the reconfigurability, as well as the PUF-to-PUF unclonability, here we study the physical unclonability of the PUFs in consecutive configuration operations. Starting from the initial state, we consecutively randomly configure $PUF_1$, and analyze the entropy, n-HD and CC metrics (Fig. 3d). As demonstrated, the entropy varies within ~0.7 to ~0.95, the CC ~ 0.2 to 0, while the n-HD remains at ~0.5, proving an almost ideal physical unclonability of the PUFs in consecutive configuration operations towards practical hardware security applications. Note that the PUFs is configured at 0.1 ms per state. The energy consumed for the configuration is estimated as ~0.432 to 34.992 fJ, given that the individual transistors require ~0.432 to 3.888 fJ to

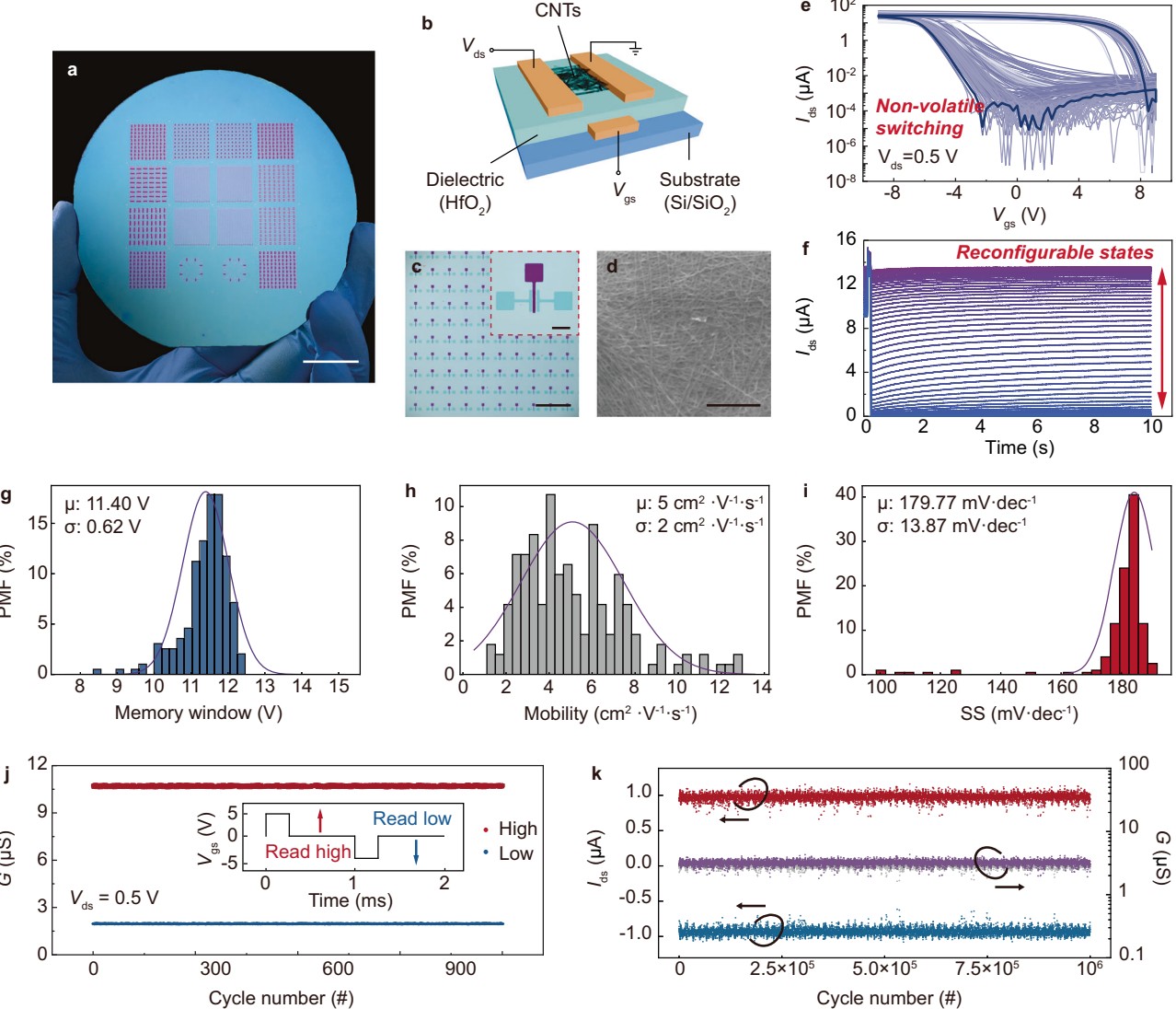

**Fig. 1 | Wafer-scale carbon nanotube charge-trapping transistors. a** Carbon nanotube charge-chapping transistors on a 4-inch wafer, with **b–d** showing the device configuration, zoomed-in device array, and scanning electron microscopic carbon nanotube thin-film. Scale bars −2 cm, 100 μm, 1 mm, and 1 μm. **e** Transfer outputs of 300 randomly sampled transistors, showing consistent yet varying non-volatile memory switching. **f** Reconfigurable memory, showing over 32 easily distinguishable stable states. Possibility mass function (PMF) of **g** the memory window, **h** the mobility extracted from the transfer outputs, and **i** the sub-subthreshold

swing, sampled from 300 random transistors across the wafer. **j** Cycling high/low conductance state configuration with 1 kHz pulsed gate signal, showing stable state configuration across 1000 cycles. **k** Cycling challenge-responses as probed by positive and negative 1 MHz pulsed drain signals at a medium conductance state, showing stable yet varying responses over $10^6$ cycles. The $10^6$ raw data points are plotted with 1000-point intervals due to the limitations of the plot drawing software.

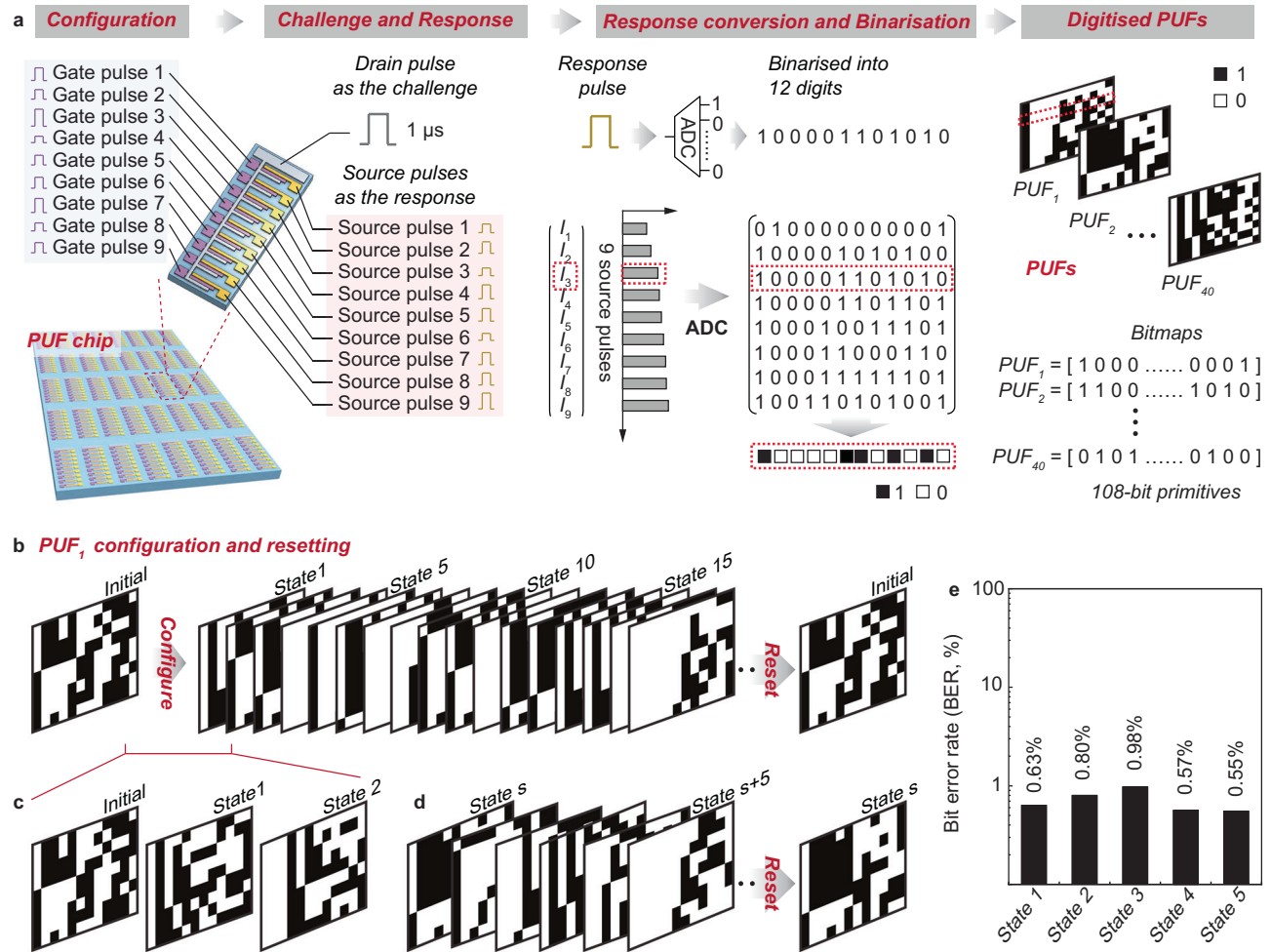

**Fig. 2 | Chip-scale PUFs and operation. a** Operation diagram of the PUFs. In the prototyped chip, 40 PUFs are fabricated, where each PUF consists of nine carbon nanotube transistors connected via a common drain. In operation, gate pulses are applied to configure the individual transistors, a 1 MHz pulsed voltage signal is applied to the drain as the challenge, and the sources output current pulses as the response that are then converted and binarized to generate the digitized PUF primitives. In binarization, each of the response pulses generates 12 binary digits via ADC. PUF primitives of a 108-bit length are generated. See Supplementary Fig. 4 for the optical microscopic images of the PUF chip, and Supplementary Fig. 6 for the hardware operation of the PUFs. **b** Configuration and resetting of the first PUF, denoted as PUF$_1$, from the initial state, with **c** zoom-in plots showing the first three consecutive states. The initial state means all the individual transistors are configured to the initial high conductance states. **d** Configuration and resetting of PUF$_1$ from a random state, s. **e** Bit error rate (BER) of our PUFs at the initial, final, and three other states. See also Supplementary Fig. 8 for the detailed BER assessment. All the BER is lower than 1%, proving the operational reliability of our PUFs.

configure. This minimal energy consumption is ideal for edge applications.

## Attacking resilience

To be applied in practical hardware security applications, the PUFs must ensure resilience to advanced machine learning and artificial intelligence (AI) attacking[16]. Advances show that regression models can be a powerful tool to attack PUFs and predict the primitives[17]. Here we adopt the *extreme gradient boosting* (XGBoost)[18] to attack the PUFs. XGBoost, with the capability designed to address structured data, is well-suited to attack PUFs[19]. As illustrated in Fig. 4a, the XGBoost is first trained in supervised learning with the primitives generated from the PUFs, and after training, the XGBoost is assigned to predict primitives. We train the XGBoost with 20,000 primitives. The results show the XGBoost proves a poor attacking to the PUFs, with an averaged prediction accuracy of only 62.61% over ten tests (Supplementary Figs. 12 and 13). To study the attacking resilience further, we analyze the n-HD and CC between the experimental and predicated primitives. As shown in Fig. 4b–e, the n-HD is ~0.4 and the CC is ~0.2, proving high-level uniqueness and irrelevance of the predicated primitives from the experimental ones, confirming the resilience of the PUFs to XGBoost

attacking. See also Supplementary Figs. 14–17 for the n-HD and CC matrix details.

Besides regression models, deep learning is widely reported successful in attacking PUFs[20,21]. Here we adopt the *generative adversarial networks* (GAN)[22] to attack the PUFs. As illustrated in Fig. 4f, the GAN consists of two deep neural networks, with one assigned as the *generator* for learning the primitives and predicating primitives, and the other assigned as the *discriminator* to distinguish the predicated primitives from the experimental ones. See also Supplementary Fig. 18 for the detailed GAN architecture. We train the *generator* with 16,000 primitives and allow the prediction of 4,000 primitives. We then train the *discriminator* to evaluate the prediction. The training loss shows that the GAN is well-trained for performing this task (Supplementary Fig. 19). The evaluation results prove again a poor attacking of the GAN to the PUFs, with an averaged prediction accuracy of only 51.31% over ten tests (Supplementary Fig. 20). We again analyze the n-HD and CC to study the attacking resilience further. As shown in Fig. 4g–j, the n-HD is ~0.5 and the CC is ~0, proving the predicated primitives are completely unique and irrelevant to the experimental ones, confirming the resilience of the PUFs to GAN attacking. See also Supplementary Figs. 21–24 for the n-HD and CC matrix details.

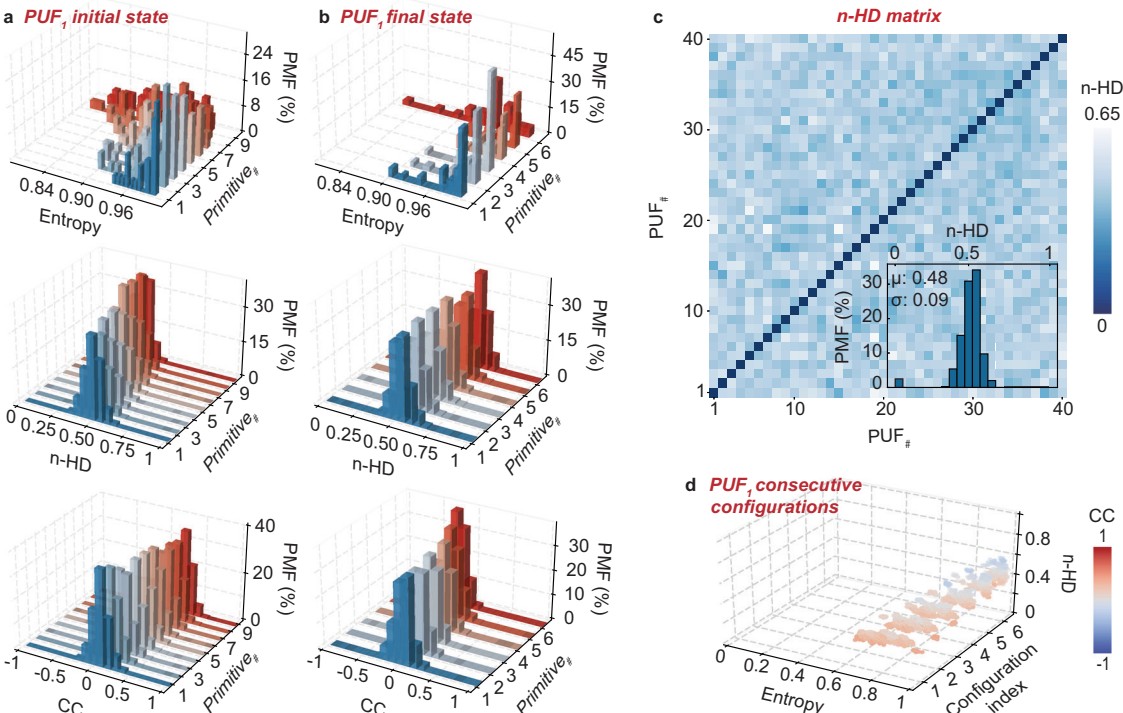

**Fig. 3 | Physical unclonability.** Entropy, normalized Hamming Distance (n-HD), and correlation coefficient (CC) of the first PUF, i.e., $PUF_1$, at **a** the initial state and **b** the final state. The final state means that all the individual transistors are configured to the final low conductance states. See Supplementary Fig. 11 for the entropy, n-HD, and CC in five other randomly configured states, where the individual transistors are randomly configured in each of the states. Entropy - 1, n-HD - 0.5, and CC - 0 in all the tests prove an ideal physical unclonability of the PUFs, i.e.,

ideal randomness, uniqueness, and irrelevance characteristics of the PUFs. **c** PUF-to-PUF n-HD matrix across the PUF chip at the initial state, with n-HD - 0.48 proving all the individual PUFs are unique from one another across the PUF chip. The insert shows the PMF of the n-HD. **d** Entropy, n-HD, and CC of $PUF_1$ in consecutive configuration operations, with entropy -0.7 to -0.95, n-HD - 0.5, CC - 0.2 to 0 proving an ideal physical unclonability. See Supplementary Table 2 for the detailed physical unclonability metric values.

The attacking resilience suggests the potential of the PUFs in securing edge devices. Here we present a common authentication solution (Fig. 5a)[23], where the central server stores the PUF primitives along with the reconfiguration and challenge instructions. In authentication, the edge user sends a request to the server and is then feedback with the instruction for generating a PUF primitive that is sent back to the server for authentication—if the primitive agrees with that stored in the server, the user is authorized as *right user* for access. Given the attacking resilience and reconfigurability of the PUFs, the authentication can achieve effective hardware security. Besides, keys may be generated with PUFs to secure data communications between the edge devices[13]. Random numbers are commonly used for key generation[24]. However, the lack of entropy renders limited randomness[25]. Here we present secure key generation using PUFs (Fig. 5b)[4]. In the simpler approach 1, at *enrolment*, the random number is encoded with the PUF primitives, such that the output engages the ideal entropy from the PUF; at *key generation*, the output through Hash function generates the key with enhanced randomness. To further enhance the randomness, at *key generation* in approach 2, the output may be encoded with the PUF primitives multiple times before Hash function. Note that the key generated can pass all the NIST SP800-22 tests, proving their high-level randomness (Supplementary Table 3). Through the PUF-based approaches, the keys carry the high-level entropy of the PUFs and ensure the security of edge communications. See Supplementary Fig. 25 for the secure data communication protocols using PUF-based private and public keys where a public-key cryptosystem Rivest Shamir Adleman (RSA) is used for the data traffic encryption[17].

Although the PUFs can secure the edge devices, with resilience against advanced machine learning and AI attacking, the

authentication and communications may still be attacked by brute force cracking (Fig. 5c), a common method to compromise the security[23]. In brute force cracking, one bit of the PUF primitives is challenged at a time, and the challenge is repeated until successful. The cracking time is estimated as the averaged time taken to crack, and is correlated to the bit length of the primitives (Fig. 5d). A 108-bit PUF primitive can require an estimated $10^{16}$ years to crack, far beyond the time limit for cracking in practical hardware security applications. On the other hand, meanwhile, the probability of achieving successful cracking in 10,000 attempts is less than $1/10^{25}$ (Fig. 5e), thereby providing high-level hardware security in practical applications. Note that the machine learning and AI attacking as well as brute force cracking resilience of the PUFs far exceeds other studies (Supplementary Table 1).

## Secure self-driving
As discussed above, our PUFs prove performance outperforming previous reports (e.g., refs. 9,17,26) in, for instance, reconfigurability, physical unclonability, and attacking resilience (Supplementary Table 1). We illustrate quantitative comparison in some of the metrics in Fig. 5h, g. This well demonstrates the capability of our PUFs in securing hardware security in edge scenarios. Here we explore the potential application in self-driving. Self-driving is a cutting-edge technological advancement that leverages artificial intelligence to deliver intelligent and secure driving experiences[27,28]. As discussed, ensuring the security of self-driving requires continuous authentications, as well as the generation and transmission of sensitive driving data between vehicles and central communication infrastructures. However, the vehicle authentication and communications are susceptible to cyber threats, including jamming, eavesdropping, and

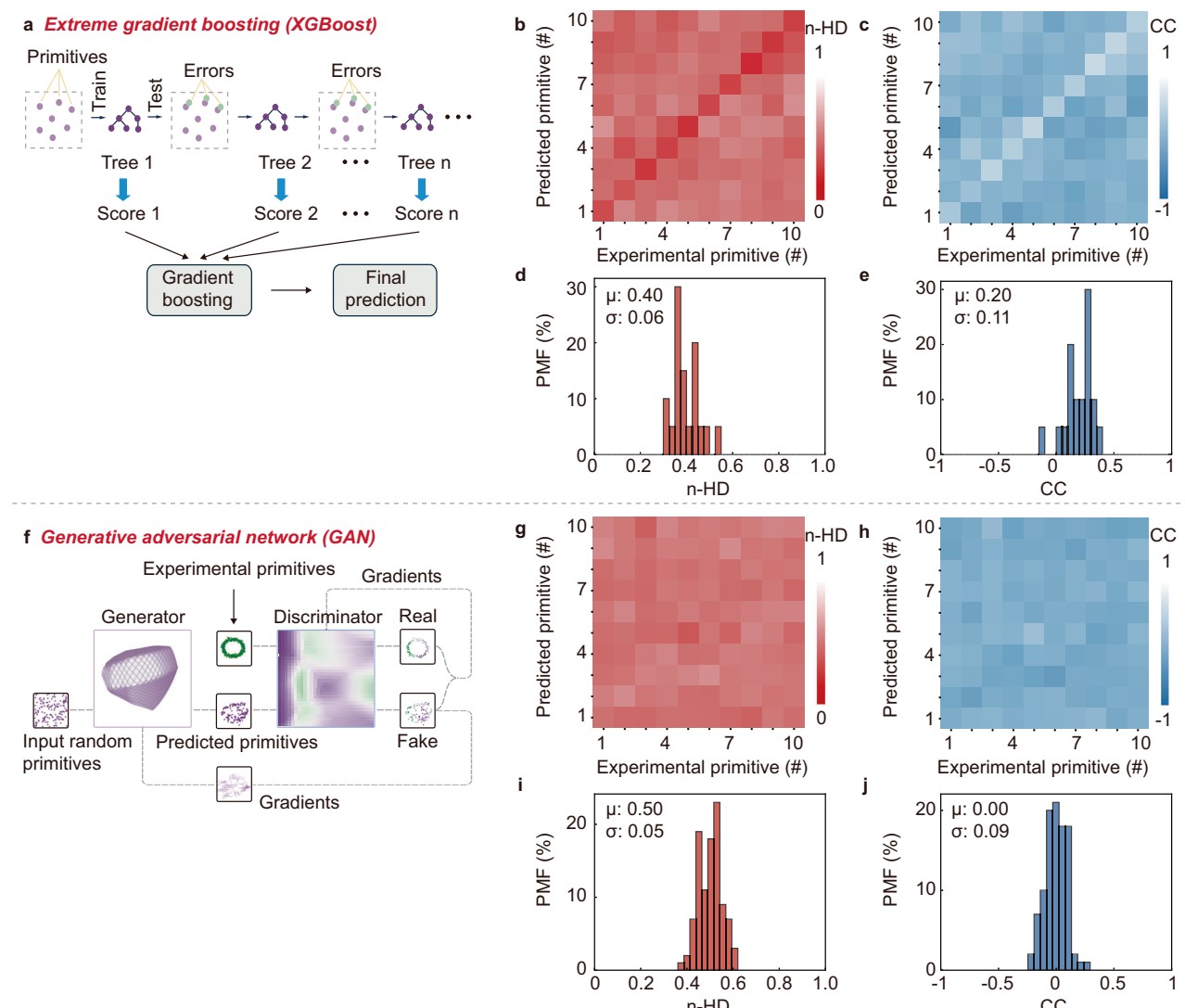

**Fig. 4 | Machine learning and AI attacking. a** Attacking by XGBoost, showing repeated attacks through gradient-boosted decision trees. Heat maps of **b** n-HD and **c** CC for the predicted primitives corresponding to the experimental primitives in XGBoost attacking, and PMF of the corresponding **d** n-HD and **e** CC. n-HD - 0.4 and CC - 0.2 prove resilience of the PUFs to XGBoost attacking. See Supplementary Figs. 14–17 for the n-HD and CC metric details. **f** Attacking by GAN, where GAN consists of two networks−the *generator* after trained with the experimental primitives generates predicated primitives, and the *discriminator* distinguishes the predicted primitives from the experimental ones. See Supplementary Fig. 18 for detailed GAN topology. Heat maps of **g** n-HD and **h** CC in GAN attacking, and PMF of the corresponding **i** n-HD and **j** CC. n-HD - 0.5 and CC - 0 prove an ideal resilience of the PUFs to GAN attacking. See Supplementary Figs. 21–24 for the n-HD and CC metric details.

spoofing, due to the use of open-wireless data exchange channels[29]. Protecting self-driving from attacks requires advanced hardware security solutions.

Here we explore self-driving vehicular network modeling with PUF-based key exchange protocols embedded to secure vehicle communication on OMNeT++ platform (Fig. 6a). OMNeT++ is a widely adopted general platform for building network simulators to implement and study real-time road traffic and vehicle communications. In this modeling, the map of Central Hong Kong is used, and the secured communication among the vehicles is in general phased into *Reconfiguration*, *Authentication*, and *Communication* stages (Supplementary Fig. 26). Briefly, *Reconfiguration* is designed to allow the vehicles to update the identities, *Authentication* is to allow the trust authority to verify the identities of the vehicles, and *Communication* is to allow the vehicles to authenticate their identities before establishing secure communication. See Supplementary Figs. 27–29 and Supplementary Note 1 for the detailed implementation processes and protocols, where PUF-based keys are exchanged among the vehicles and the trust

authority for the mutual-verification and authentication handshakes. Particularly, in this modeling, PUF is embedded a *PUF Module* in each vehicle module to 108-bit responses upon challenges. This *PUF Module* is allowed to feature the characteristics of our carbon nanotube PUFs including the reconfigurability and input-output response times. As such, a time delay of -1 ms is cost by the *PUF Module* in generating the 108-bit responses. Note that a lightweight public-key cryptosystem Elliptic Curve Cryptography[29] is used here for PUF-based key exchange at the edge scenario.

In this self-driving vehicular network modeling, self-driving networks of 10−100 vehicles are implemented. See Fig. 6b and Supplementary Movie 1 for the real-time self-driving of 100 vehicles. The results prove smooth road trafficking flows and vehicle authentication and communications. To detailly evaluate the self-driving trafficking, we study the time delay, data transmission, and computation cost due to the vehicle authentication and communications. As demonstrated, the time delay is typically -12 ms for the 10−100 vehicle network scales (Fig. 6c−f), with a maximum delay of -100 ms (Fig. 6g). This well meets

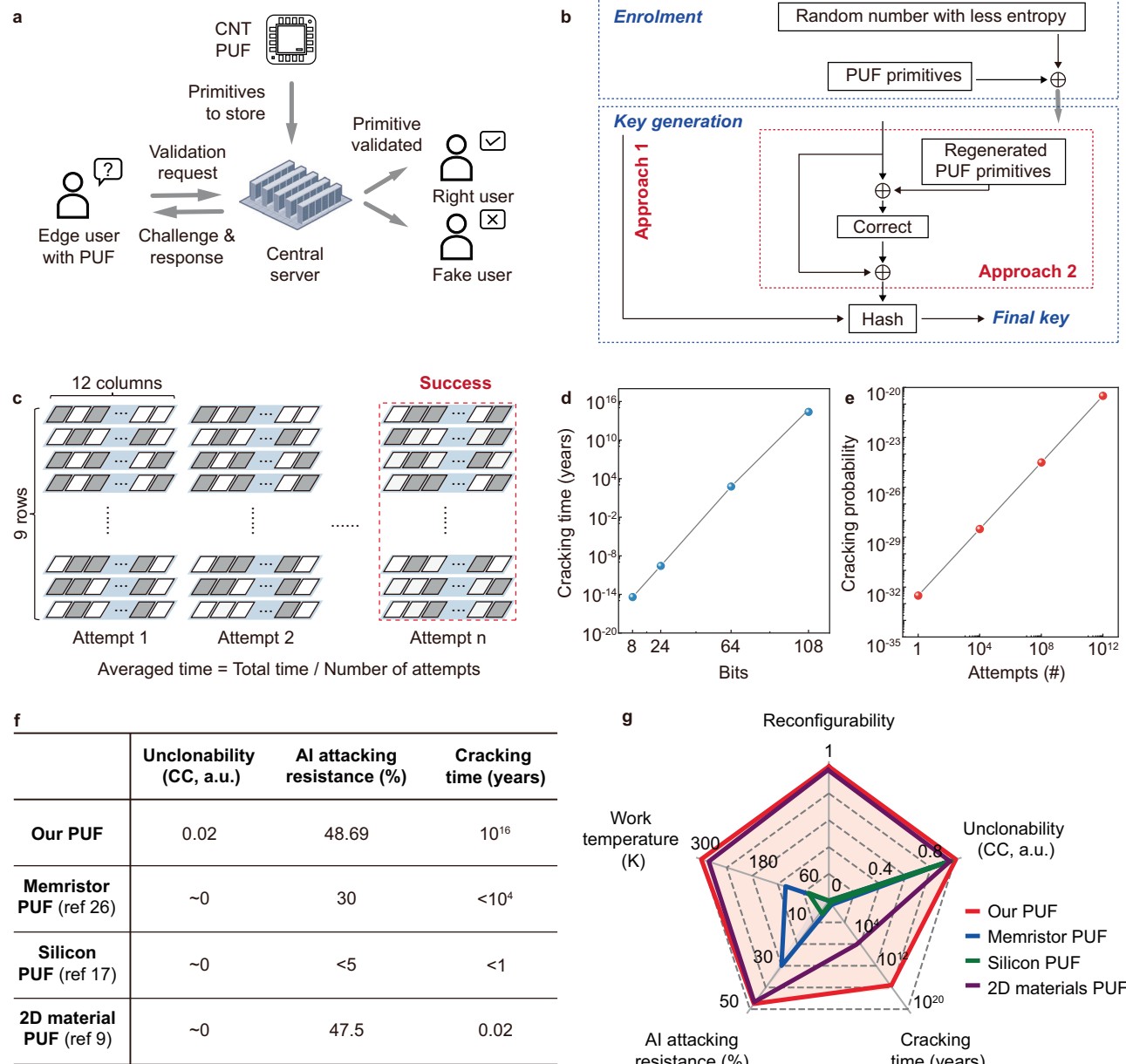

**Fig. 5 | Brute force cracking. a** Authentication with PUFs – PUF generates and stores primitives in the central server; the edge user with PUF sends a validation request to the server, and is then feedback with the specific configuration and challenge to generate a primitive to be sent back to the server; if the primitive agrees with the stored one, the edge user is authorized as *right user*. **b** Key generation with the PUFs, with *enrolment* and *key generation* stages—*enrolment* engages the random number with the entropy of the PUFs; *key generation* generates the key via two approaches, with the simpler approach generating the key directly through Hash function, while the other complex approach further engaging the

entropy of the PUFs before key generation through Hash function. **c** Brute force cracking of the PUFs, with the primitives challenged by one bit at a time until successful. The averaged cracking time is estimated by the total time divided by the number of cracking attempts. **d** Plot of the cracking time versus the bit length of the primitives, and **e** plot of the cracking probability versus the number of attempts, showing the primitives of a 108-bit length take an estimated $10^{16}$ years to crack. **f** Quantitative comparison and **g** the radar plot in some key performance metrics of our PUFs with state-of-the-art reports. See Supplementary Table 1 for a detailed quantitative comparison.

the self-driving requirements $(50\,ms)^{30}$. This well-acceptable time delay partly arises from the fast generation of primitives of the PUFs (~1 μs). On the other hand, the data transmission for a communication process is estimated 324 bits per vehicle, and the overhead for authentication is 1544 bits per vehicle (Fig. 6h). The associated computation cost is estimated as ~0.87 ms, given that each computation goes through six Hash function operations, two symmetric encryption and decryption operations, four elliptic curve point addition and four elliptic curve scalar multiplication operations, and two PUF operations in the authentication and communications. This data transmission and computation cost overhead outperform the other studies[31–33] and,

importantly, well meet the self-driving requirements[30]. The results prove the successful use of the PUFs in enabling low-delay, lightweight vehicle authentication and communications in self-driving. In summary, our PUF-based key exchange protocol achieves a typical time delay of ~12 ms for the networks and enables lightweight authentication and communication with only 324 bits per vehicle for communication and 1544 bits for authentication, alongside a low computational cost of ~0.87 ms per vehicle. As previously demonstrated, our PUFs can provide high-level resilience to advanced machine learning and AI attacking as well as brute force cracking. Given this, as the success possibility of cracking in 10,000 attempts is

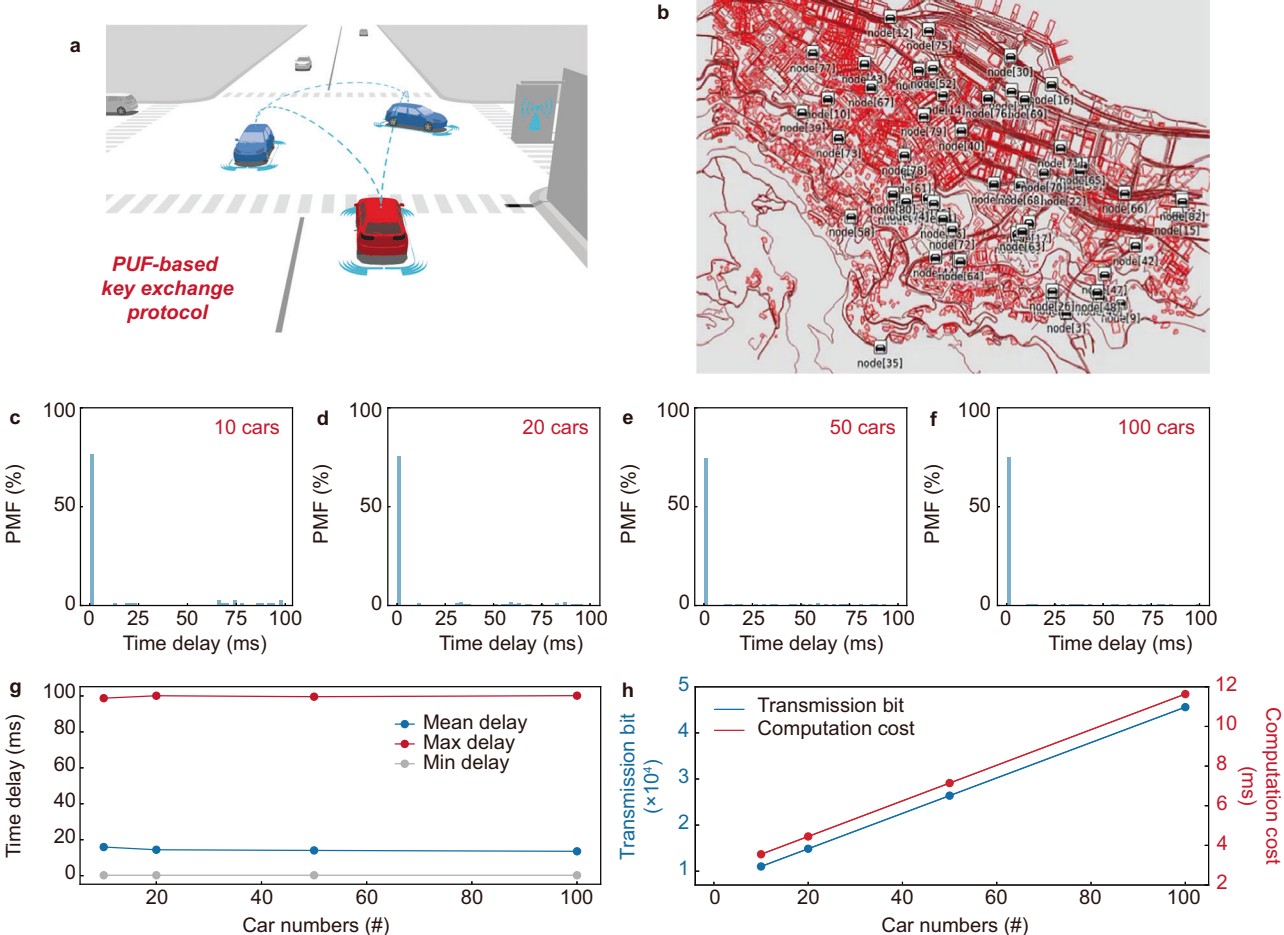

**Fig. 6 | Secure self-driving. a** Vehicle authentication and communications using a PUF-based key exchange protocol. See Supplementary Fig. 26 for the detailed implementation scheme of the vehicular network, and Supplementary Figs. 27–29 for the PUF-based key exchange protocol implementations. **b** Self-driving modeling of 100 vehicle network in Central Hong Kong. The modeling is conducted on OMNeT++. See Supplementary Movie 1 for the real-time self-driving. **c**–**f** PMF of time delays resulted from the vehicle authentication and communications for self-driving networks of 10–100 vehicles, showing that the PUF-based data exchange protocol ensures negligible time delays. **g** Plot of the time delays versus the self-driving vehicle numbers, showing a mean delay of ~12 ms, ideal for real-world self-driving security applications. **h** Plot of the data transmitted between the vehicles and the associated computation cost versus the self-driving vehicle numbers, proving balanced hardware security, data transmission bits, and computation cost.

less than $1/10^{25}$, it is nearly impossible to attack successfully in practical self-driving in a time window of ~10 ms.

## Discussion

We have developed chip-scale reconfigurable PUFs to enhance hardware security in edge intelligence. The PUFs, designed with carbon nanotube based charge-trapping transistors, achieve an exceptional reconfigurability exceeding $10^{13}$ states by exploiting the non-volatile charge-trapping memory, and prove physical unclonability with ideal randomness, uniqueness, and irrelevance by harnessing the entropy variation arising from the randomly arranged carbon nanotube network. With the physical unclonability and reconfigurability, the PUFs demonstrate robust resilience against advanced machine learning and AI attacking as well as brute force cracking. These performance metrics surpass both existing hardware security benchmarks and state-of-the-art advancements, manifesting the potential of the PUFs for securing edge intelligence applications. As a practical demonstration, we show that the PUFs can be embedded to enable secure self-driving with low-delay, lightweight secure vehicle authentication and communications.

Given this performance, along with their scalability and low-power operation as well as cryogenic temperature robustness, the PUFs are readily integrated into diverse edge intelligence systems beyond self-driving vehicles, for instance, robotics, drones and unmanned aerial vehicles, and IoT systems, for hardware security. Looking forward to bridging the lab-to-fab gap for our PUFs towards the real-world applications, our PUFs expect to adapt to industrial-scale advanced photolithographic systems for the manufacture of PUFs with high density of miniaturized transistors and low power consumption. The manufacture also expects to meet CMOS integration towards integral PUFs with the analog front end and digital logic fabricated and integrated onto one single chip. At the system level, multi-channel pipe-lined ADCs with on-chip buffering and microprocessors may be integrated for operating the PUFs. Furthermore, hardware-algorithm co-designs in operation instructions and error correction strategies are demanded for practical operations of the PUFs.

## Methods

### Carbon nanotubes

Carbon nanotubes are processed following our previous report (12). 1 mg/mL single wall carbon nanotubes are dispersed in toluene with 2 mg/mL poly[N-(1-octylnonyl)-9H-carbazole-2,7-diyl] (PCz). The mixture is tip-sonicated (400 W, 1 h) and centrifuged (20,000 × g, 2 h), and 90% of the supernatant is collected. The collected supernatant is vacuum filtered, washed with tetrahydrofuran, and redispersed in chloroform with an additional sonication (400 W, 0.5 h) to obtain high-purity semiconducting-phase carbon nanotubes in solution. The

solution is deposited to yield carbon nanotube thin-films for PUF fabrication by dip coating (lowering speed 500 μm/s, lifting speed 100 μm/s, repeated for 15 times), followed by post-bake (150 °C).

## PUF fabrication

The PUFs are fabricated via photolithography. The $SiO_2/Si$ substrate is cleaned with deionized water, acetone, and isopropanol via bath sonication for 10 min each. The carbon nanotube channel is patterned with photolithography and etched with oxygen plasma, and is then washed with acetone to remove the photoresist. The source, drain, and gate electrodes (5 nm Ti and 15 nm Au) are patterned with photolithography, deposited via electron beam evaporation (IVS EB-600), and finalized with a lift-off process. The 20 nm $HfO_2$ dielectric layer is deposited by atomic layer deposition. The fabricated PUFs are then baked at 150 °C for 1 h.

## Transistor characterizations

The output and transfer of the transistors are measured with Tektronix Keithley 4200A-SCS in DC mode. The multistate conductance states and cycling endurance of the transistors are measured with an arbitrary waveform generator (Siglent SDG7032A) and a digital storage oscilloscope (SDS2354X), in conjunction with operational amplifiers (TL082CP) and external resistors that match the signals. The state configuration is conducted with the pulse measure unit (PMU) of Tektronix Keithley 4200A-SCS. All the above tests are performed in ambient conditions. The 100–400 K tests are performed in high vacuum (~$10^{-6}$ mbar), and the temperature of the probe stage is regulated by a thermostat with a heating plate and liquid nitrogen cooling.

## PUF operation

For the initial state PUF test, the test is conducted when the individual transistors are all configured to the initial high conductance states. For the other reconfigurable state tests, the states of the individual transistors are configured as designed with PMU. Primitive generation is performed with an ADC testing board. The ADC testing board consists of analog-to-digital converter (Zhengzhou Hengkai Electronic Technology Co.) modules. The PUFs are connected to the probe station, the arbitrary waveform generator (Siglent SDG7032A), the digital storage oscilloscope (SDS2354X), the ADC testing board, and also transimpedance amplifiers (TIA) via a breadboard, as shown in Supplementary Fig. 6. The transistors are configured and reset (via the gates) and applied with the challenge pulse (via the common drain) to get the current pulses as the response (via the sources). Each of the response pulses from the sources is converted to a voltage pulse via the TIA modules, and then binarized into 12 binary digit bits via the ADC modules. The output binary digit bits are acquired via a laptop port using a specialized acquisition software of the ADC modules. The PUFs are operated with a frequency of 10 kHz for the state configuration, and a frequency of 1 MHz for challenge-response generation. However, due to the noise and frequency limitations of the breadboard and the ADC testing board, the actual operation frequencies may be decreased a bit.

## XGBoost attacking

20,000 primitives containing 108-bit binary digits are separated into 10,000 odd rows and 10,000 even rows by parity. They are then reorganized into 216 columns of data frames from bit 1 to bit 108. By doing so, the primitives are restructured such that one column is an odd row containing 108 "feature" columns, and another column is an even row containing 108 "target" columns. The data frames are split, with 80% for training and 20% for validation. Following this, XGBoost classification (https://xgboost.readthedocs.io/en/stable/#) is used to fit a specific XGBoost classifier for each primitive bit by looping through all the 108 bits. After each bit model is trained, the accuracy of

the validation set is evaluated, and the predictions are recorded and collected. In this work, n-HD and CC between 10 predicted primitives and 10 experimental primitives are extracted to draw the heat maps. The above XGBoost attacking is repeated ten times, with the data frames randomly split in each of the tests.

## GAN attacking

20,000 primitives are converted into a tensor by replacing 0 with −1 as the dataset. The dataset is then split, with 80% for training and 20% for validation. Following this, GAN (https://github.com/goodfeli/adversarial) is adapted for modeling the primitives. GAN consists of a *generator* and a *discriminator*. For the generator, the size of each layer is increased sequentially, such that the generator can map random input to PUF primitives. For the discriminator, it handles the input primitives to determine if the primitives are from the dataset. The discriminator uses a *Leaky Rectified Linear Unit* (ReLU) as the activation function, and *Wasserstein* loss formulation plus a gradient norm penalty to stabilize the training. By doing so, the generator keeps generating primitives, and the discriminator keeps learning the primitive generation in the dataset. The training is repeated for 150 epochs. After the training, the accuracy of the discriminator on validation is evaluated. In this work, n-HD and CC between 10 predicted primitives and 10 experimental primitives are extracted to draw the heat maps. The above GAN attacking is repeated ten times, with the 20,000 primitives randomly split in each of the tests.

## Self-driving

Self-driving is modeled on OMNeT++ 6.0.2 (https://omnetpp.org/). First, the map of Central Hong Kong (114°15′ E, 22°29′ N to 114°17′ E, 22°27′ N) is downloaded from OpenStreetMap (https://www.openstreetmap.org/), a public street map website. Then, the files containing the map and 10-100 vehicle routing are built in the Simulation of Urban Mobility 2.10.0 (SUMO). Finally, self-driving modeling runs on the OMNeT++ platform using SUMO in Linux. OMNeT++ is adapted for detailed packet-level simulation of data transmission and reception between the modeled self-driving vehicles, with the PUF-based key exchange protocol embedded for securing the vehicle authentication and communications. See Supplementary Fig. 26 and Supplementary Note 1 for the detailed secure communication implementation scheme. See Supplementary Figs. 27–29 for the detailed PUF-based key exchange protocol and implementation. See Supplementary Movie 1 for the real-time self-driving network of 100 vehicles.

## Data availability

All data supporting the findings of this study are available on Figshare https://doi.org/10.6084/m9.figshare.28953254.

## Code availability

Customized Python codes and C++ codes for XGBoost attacking and GAN attacking as well as PUF enabled vehicle communication on open-source OMNeT++ used in this study are available at Code Ocean https://codeocean.com/capsule/4214745/tree/v1.

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

## Acknowledgements

G.H.H. acknowledges support from RGC (24200521) and CUHK (IDB-F25ENG05), Y.L. from SHIAE (RNE-p3-21), J.F.P. and Y.Y.W. from RGC (24200521), T.M. from RGC (15306824) and ITC (ITS/150/23FP), and X.L.C. from Shenzhen Excellent Youth Program (RCYX20221008092900001). We thank Chen et al. for the XGBoost model (https://xgboost.readthedocs.io/en/stable/#), Goodfellow et al. for the GAN model (https://github.com/goodfeli/adversarial), and Kahng et al. for the GAN Lab (https://poloclub.github.io/ganlab/). The schematic illustrations including the transistor architecture, the PUF chips, the central server, and the vehicle communication process in Figs. 1b, 2a, 5a, and 6a are created using Blender under a GNU General Public License (https://www.blender.org/support/faq/).

## Author contributions

Y.L., J.F.P., G.H.H. designed the experiments. Y.L., J.F.P., Y.Y.W., L.K.S., S.W.L., P.Y.L., W.Y.C., Z.H.L. performed the experiments. Y.L., J.F.P., T.M., X.L.C., G.H.H. discussed and analyzed the data. Y.L., J.F.P., G.H.H. prepared the figures. Y.L., J.F.P., G.H.H. wrote the manuscript. All authors discussed the results from the experiments and commented on the manuscript.

## Competing interests

The authors declare no competing interests.
