## [Transparent Peer Review file · Nature Communications]

Chip-scale reconfigurable carbon nanotube physical unclonable functions

Corresponding Author: Professor Guohua Hu

Version 0:

Reviewer comments:

Reviewer #2

(Remarks to the Author)

In this manuscript, Liu et al. present chip-scale reconfigurable PUFs fabricated with carbon nanotube charge-trapping transistors. The authors report that the PUFs can achieve more than 10^{13} reconfigurable states with near-ideal physical unclonability. Besides, the authors demonstrate that the PUFs prove high-level attacking resilience to advanced machine learning and artificial intelligent attacks, including XGBoost (limiting success to 62.61%) and GAN (limiting success to 51.31%), as well as robustness against brute force cracking (requiring 10^{16} years to compromise). The above impressive performance makes the reconfigurable PUFs possible and promising to be integrated into edge intelligent systems to enhance hardware security. Finally, interestingly, as a practical demonstration, the authors model secure authentications and communications in self-driving cars in Central, Hong Kong. The results are interesting and impressive, outperforming other reports on PUFs. Particularly, the demonstrated reconfigurability in the PUFs is important for edge and new applications of PUFs. Overall, in my opinion, it is an interesting work, especially the reconfigurability of the PUF.

Before publication recommendation, this manuscript can be improved by addressing my below comments:

- 1) As demonstrated by the authors, the PUFs based on solution-processed carbon nanotubes exhibit reconfigurable states with near-ideal physical unclonability. How about the error rate and reliability for operating the PUFs at the initial state and other reconfigurable states? Once the PUFs are reconfigured, can these reconfigured states be stable in terms of the error rate?
- 2) In terms of the bit length of PUFs, a 12-bit ADC is used to convert the current values into digital bits. This produces 108 bit-length PUFs, as designed by the ADC circuit. Please explain in detail the working principle of the bit generation and PUF operation using the ADC circuit board. Can PUFs in other bit-lengths be generated?
- 3) Following my previous comment, in actual operation occasions, what would be the real bit-length of the PUF extracted cryptographic key considering the error rate? Because the extracted key length can be reduced due to the unreliability of the PUF response.
- 4) A stable digital bit generation is critical for PUF operations. In actual operation occasions, is the digital bit generation stable using the ADC circuit? Will and how the current values impact the digital bit generation? If the digital bit generation using the ADC circuit is not stable, how would the authors address this problem, that is, how to make the analog-to-digital conversion more robust through circuit or algorithm designs?
- 5) PUFs are a promising hardware security solution. Looking forward to future practical applications, since the method as presented by the authors is potentially scalable, how can the method be further improved to promote its real-world implementation and industrialization process? The authors may want to elaborate on this for a larger impact of the work.

Remarks on code availability:

The code is enough and sufficient. I can see the results on Code Ocean and they are right to me.

(Remarks on code availability)

N/A

Reviewer #3

(Remarks to the Author)

In this work, Liu et al. present the development of physical unclonable functions (PUFs) based on reconfigurable carbon nanotube (CNT) charge-trapping transistors. The fabrication and characterization of the transistors are thoroughly described, demonstrating significant potential for PUF development. After that, the authors evaluate the resilience of the developed PUFs against attacks. While they claim that the high entropy of the PUFs ensures the security of edge communications, this aspect of the work remains unclear.

The authors also include a simulation-based application within a Vehicular Ad-Hoc Network (VANET) using the OMNeT++ platform. Although this scenario is interesting, it does not clearly demonstrate how the proposed PUFs enhance robustness or privacy compared to existing solutions.

In summary, the generation of PUFs using reconfigurable CNT charge-trapping transistors is promising and shows strong potential for hardware-based security solutions. However, the practical application discussed in this work is somewhat ambiguous and should be clarified.

Specific comments to be addressed:

- The process used to configure and reset individual transistors should be explained in more detail, with experimental results to ensure reproducibility.
- The authors claim that the proposed PUFs outperform those in previous studies (Fig. S1). However, a detailed, quantitative comparison should be added in the main manuscript, including benchmarking metrics for the different key parameters.
- The sentence "PUF all prove a stable operation after even 120 days in ambient conditions" is unclear. I assume the authors mean that the devices were tested up to 120 days and demonstrated stable operation throughout that period, although the stability may extend beyond that. I recommend rephrasing the sentence to clarify this point.
- Line 120: Sentence "the primitives all present an n-HD of ~ 0.5 ", is imprecise. Please provided the obtained values. Same for entropy and CC values.
- It would be interesting to also include in Table S1 some PUF based in memristors, as [1].
- The application in the vehicular network should be more thoroughly introduced, with a clearer explanation of how the proposed technology improves upon the current state-of-the-art, including quantifiable metrics.
- In line 212, the authors affirm that they have adapted "a PUF-based key exchange protocol, following authentication and key generation in Fig. 5". As the PUF is being used to generate cryptographic keys, it would be convenient to study if the proposed PUFs primitives pass the NIST SP800-22 tests [2].
- Regarding the cryptosystems used in the example, Figure S21 presents RSA as the public-key cryptosystem used in the example, while in Figures S22 to S24, Elliptic Curve Cryptography (ECC) is used for the implementation of the protocols. This should be clarified. In addition, a reference to the article/ articles where these authentication protocols are presented is required.
- The discussion of how PUFs integrate into common system architectures (such as in Fig. S21) should be expanded, indicating the implications and potential limitations.
- The term "hash function" is misspelled in several places (e.g., page 5, line 187).

References

- [1] Ibrahim, Hebatallah M., Heorhii Skovorodnikov, and Hoda Alkhzaimi. "Resilience evaluation of memristor based PUF against machine learning attacks." *Scientific reports* 14.1 (2024): 23962.
- [2] Pareschi, Fabio, Riccardo Rovatti, and Gianluca Setti. "On statistical tests for randomness included in the NIST SP800-22 test suite and based on the binomial distribution." *IEEE Transactions on Information Forensics and Security* 7.2 (2012): 491-505.

(Remarks on code availability)

I reviewed the code but did not run it, as authentication was required on the platform where the authors uploaded the code, which could compromise the reviewers' anonymity. In any case, the information provided appears to allow the application to be installed and run on a self-managed computer.

Reviewer #4

(Remarks to the Author)

(Remarks on code availability)

Version 1:

Reviewer comments:

Reviewer #2

(Remarks to the Author)

Thank you for thoroughly addressing my concerns about reliability through detailed and extensive experiments, as well as the length of the cryptographic keys. I have no further concerns.

(Remarks on code availability)

Reviewer #3

(Remarks to the Author)

The authors have addressed all my comments and suggestions and the manuscript has been significantly improved. I think it can be accepted in its current form.

(Remarks on code availability)

The code allows to reproduce the results presented in the paper from the data provided by the authors.

Reviewer #4

(Remarks to the Author)

(Remarks on code availability)

Point-by-point response to the reviewers' comments

Reviewer #2 (Remarks to the Author):

In this manuscript, Liu et al. present chip-scale reconfigurable PUFs fabricated with carbon nanotube charge-trapping transistors. The authors report that the PUFs can achieve more than 10^{13} reconfigurable states with near-ideal physical unclonability. Besides, the authors demonstrate that the PUFs prove high-level attacking resilience to advanced machine learning and artificial intelligent attacks, including XGBoost (limiting success to 62.61%) and GAN (limiting success to 51.31%), as well as robustness against brute force cracking (requiring 1016 years to compromise). The above impressive performance makes the reconfigurable PUFs possible and promising to be integrated into edge intelligent systems to enhance hardware security. Finally, interestingly, as a practical demonstration, the authors model secure authentications and communications in self-driving cars in Central, Hong Kong. The results are interesting and impressive, outperforming other reports on PUFs. Particularly, the demonstrated reconfigurability in the PUFs is important for edge and new applications of PUFs. Overall, in my opinion, it is an interesting work, especially the reconfigurability of the PUF.

Before publication recommendation, this manuscript can be improved by addressing my below comments:

We thank the reviewer for the favorable comments on our work. Here we address the queries raised in our responses below point by point:

1) As demonstrated by the authors, the PUFs based on solution-processed carbon nanotubes exhibit reconfigurable states with near-ideal physical unclonability. How about the error rate and reliability for operating the PUFs at the initial state and other reconfigurable states? Once the PUFs are reconfigured, can these reconfigured states be stable in terms of the error rate?

We have conducted rigorous tests to study the bit error rate and reliability for operating our PUFs at the initial and other reconfigurable states, including 1) test on the current outputs I_{out} from the 9 individual transistors of a PUF (Fig. R1), and 2) test on the bit error rate of the PUF (Fig. R2).

- *Test on current outputs I_{out}*

To operate our PUFs (Fig. 2), the first step is to reset and configure a PUF to a state with voltage

pulses V_{gs} applied onto the gates of the 9 individual transistors of the PUF; the second step is to challenge the PUF with a voltage pulse V_{in} applied onto the common drain of the PUF, at the time V_{gs} keeps at 0 V; and the third step is to convert the current outputs I_{out} read from the sources of these 9 transistors into 12-bit digit numbers. Therefore, the reliability for operating these 9 transistors determines the bit error rate and reliability for operating our PUFs.

In our test (Fig. R1a), we configure a typical PUF at the initial state, the final state, and three other states in between. The initial state means that all the 9 transistors are configured to the initial high conductance states, i.e. $V_{gs} = 0$ V. The final state means all the transistors are configured to the final low conductance states, i.e. $V_{gs} = -9$ V. In the three other states, all the transistors are configured with $V_{gs} = -3$ V, -5 V, and -7 V, respectively. After configuration, in these 5 states, we challenge the PUF with 0.5V DC V_{in} for 100,000 sampling times as the challenge, and read the current outputs I_{out} from these 9 transistors. We choose this method to directly read the transistor responses under the same voltage stimulus and it ensures the stability of the input voltage, thereby avoiding any impact on the accuracy of the results. Each of the current outputs I_{out} in Fig. R1b corresponds to 100,000 response data points.

As demonstrated by the current output I_{out} (Fig. R1b), all the 9 transistors in these 5 states behave highly stable through the 100,000 challenge times. Note that these 9 transistors in the PUF show differences in the conductance levels, arising from the device variations. Nevertheless, the highly stable performance of the 9 transistors in these 5 different states proves the reliability of their operation and, importantly, guarantees a reliable operation of the PUF.

Figure R1. Current outputs from the 9 individual carbon nanotube transistors in a typical PUF through 100,000 challenge times. (a) Circuit diagram showing the test. (b) Current outputs I_{out} from the 9 transistors in the PUF. All the 9 individual transistors are configured at

the initial, final, and three other states by gate pulses V_{gs} . V_{gs} is 0 V, -3 V, -5 V, -7 V, and -9 V to configure the State 1-5, respectively. Current outputs I_{out} are probed under the challenge, i.e. drain V_{in} with amplitude 0.5 V DC is composed of 100,000 sampling times. All the 9 transistors in these 5 states give highly stable outputs through the 100,000 challenge times.

- **Test on bit error rate**

In the test, the current outputs I_{out} are converted into binary digit numbers, and each of the current pulses produces a 12-bit digit number. The bit error rate (BER) is computed by dividing the number of error bits by the total number of bits (Zhong, Donglai, et al. *Nature Electronics* 5.7 (2022): 424-432). To rigorously test the BER, we calculate the BER from all the 9 individual transistors in these 5 states when performing the analog-digital conversion (Fig. R2a), and the averaged BER of the PUF in these 5 states (Fig. R2b). As shown, the individual transistors give a BER of <2 % in all these 5 states. This leads to a BER of the PUF varying from 0.55% to 0.98%, suppressing state-of-the-art reports (e.g., 1.7% in Shao, Hanyong, et al. 2023 *International Electron Devices Meeting (IEDM). IEEE, 2023*). This proves the reliability of operating our PUF.

Figure R2. Bit error rate (BER) of operating our PUFs. (a) BER of all the 9 individual transistors of a typical PUF in the 5 states. The BER is <2% for all the transistors in all the states. (b) BER of the PUF in the 5 states. The BER is <1% for all the states.

To clarify and address the possible queries from the readers about the BER and reliability for operating our PUFs, we have now included the following discussion and Fig. R1 and R2 in the revised manuscript and supporting information:

(Line 108) See also Fig. S7 for the highly stable challenge-response generation from the transistors across 100,000 sampling times in the initial and other states. This stable transistor operation can give rise to a limited bit error rate (BER) in PUF primitive generation. Indeed, we demonstrate that the BER in practical operation of our PUFs is <1% (Fig. 2e, Fig. S8).

2) In terms of the bit length of PUFs, a 12-bit ADC is used to convert the current values into digital bits. This produces 108 bit-length PUFs, as designed by the ADC circuit. Please explain in detail the working principle of the bit generation and PUF operation using the ADC circuit board. Can PUFs in other bit-lengths be generated?

As discussed above, to operate our PUFs (Fig. 2), after configuration and the challenge, analog current pulses I_{out} are read as the responses from each of the 9 individual transistors for analog-digital conversion using ADC.

We design an ADC testing board to perform the analog-digital conversion. As illustrated in Fig. S6, each individual transistor of a PUF is connected to an ADC via a trans-impedance amplifier (TIA). To perform the conversion, the analog current pulses I_{out} are first converted into analog voltage pulses using the TIA and then binarized into digit numbers using the ADC. The ADC (supplied by Zhengzhou Hengkai Electronic Technology Co.) can output a 12-bit digit number from each individual analog voltage pulse read from the TIA. Given that there are 9 parallel analog current pulses I_{out} in each challenge, a 108-bit PUF primitive can be generated.

As the default function of the ADC we use, each analog voltage signal (in the range of 0 – 5 V) to the ADC can be converted into digit numbers ranging from 1 to 4096, i.e., 2^0 to 2^{12} . In our practical PUF operation, the analog current pulses I_{out} as the responses to a challenge of 0.5 V pulse are dispersed in the range of $\sim 0.08 - 41.5 \mu\text{A}$, giving analog voltage pulses in the range of 0.01 – 5 V to be binarized via the ADC. This range is well within the operational range of the ADC.

PUFs in other bit lengths can also be generated. The bit length of a PUF can be designed to suit the actual requirements of practical PUF applications, and can be determined by:

$$\text{Bit length} = \text{number of devices in a PUF} \times \text{ADC bit count}$$

Therefore, we can realize different bit lengths by 1) designing and fabricating PUFs integrating other numbers of transistors, and 2) using ADC in other models to achieve other bit-length digit number generation.

To clarify and address the possible queries from the readers regarding the bit number generation principle, we have now made the following revision in the revised manuscript:

(Line 98) Note the 12-bit length analogue-digital conversion is defined by the ADC used in our

work, and other bit length digits can be generated using other ADC models.

3) Following my previous comment, in actual operation occasions, what would be the real bit-length of the PUF extracted cryptographic key considering the error rate? Because the extracted key length can be reduced due to the unreliability of the PUF response.

Yes, as pointed out by the reviewer, the real bit length of the PUF-extracted cryptographic key in actual operation occasions can be impacted by the bit error rate (BER). The real bit length can be calculated based on the correct bits excluding the error bits, as expressed by:

$$\text{Real bit length} = \text{number of total bit length} \times (100\% - \text{BER})$$

In our case, as we prove in Fig. R1 and R2, our PUFs demonstrate a highly reliable operation, with a BER of <1%. Given that 108-bit digit numbers can be generated at each of the challenges, the real bit length of our PUFs is estimated as 106-bit.

To clarify and give the readers further information on the real bit length of our PUF, we have now made the following revision in the revised manuscript:

(Line 111) Given this, the real bit length of our PUF primitives is 106-bit (i.e. $108 \times 99\%$).

4) A stable digital bit generation is critical for PUF operations. In actual operation occasions, is the digital bit generation stable using the ADC circuit? Will and how the current values impact the digital bit generation? If the digital bit generation using the ADC circuit is not stable, how would the authors address this problem, that is, how to make the analog-to-digital conversion more robust through circuit or algorithm designs?

Yes, the digital bit generation in practical operation of our PUFs is stable. As proved in Fig. R1 and R2, our PUFs demonstrate a highly reliable operation, and the BER is smaller than 1% in the initial and other reconfigured states.

As discussed, as the default function of ADC, the voltage signals input to the ADC are converted into digit numbers depending on the signal amplitudes and the operational range of the ADC. When the voltage signals drift too low, ADC operation is driven into a low-end noise-floor region, where the analog-to-digital conversion can be highly impacted by the noise and leads to error bits. When the voltage signals drift too high, ADC operation is driven into its clipping zone, where the analog-to-digital conversion can saturate thus leading to error bits.

In our practical PUF operation, the ADC we use is operated in the range of 0 – 5 V, and the

analog voltage pulses input to the ADC (converted and amplified by TIAs from the analog current pulses I_{out}) are dispersed in the range of 0.01 – 5 V. Therefore, the input to the ADC lies in the normal operational region. This ensures that the analog-to-digital conversion is robust with a limited BER.

Though analog-to-digital conversion of our PUFs is robust, further hardware and algorithm measures can be adopted to ensure its robust operation in real-world applications:

- ***Hardware measure***

Generally, in our understanding, there are two convenient approaches to further improve the operational reliability: 1) using ADC that can perform analog-to-digital conversion in a higher precision, e.g. *ADS1282 High-Resolution Analog-To-Digital Converter from Texas Instruments Inc.*, and 2) using FPGA or microprocessors instead of ADC to generate digit numbers with a higher accuracy and precision.

- ***Algorithm measure***

Algorithms may also be designed to enhance operational reliability and error bit correction. Here we list three common approaches in Fig. R3: 1) *Temporal majority voter (TMV)* to enhance the PUF operational reliability by repeatedly measuring the analog current pulses, 2) *Median value by multiple measurement* to check if the PUF response lies within the acceptable bounds, and 3) *Error-correcting code* to decode the noisy PUF response using helper data and as such, produce reliable response.

a **Temporal majority voter (TMV)**

b **Median value by multiple measurement**

c **Error-correcting code (Bose-Chaudhuri-Hocquenghem code as an example)**

Figure R3. Algorithm improvements for our PUF operation. (a) *Temporal majority voter (TMV)*, repeatedly measuring the response and selecting the most frequent output as the result. (b) *Median value by multiple measurement*, checking if the measured PUF response lies within acceptable bounds and using the median value as the result. (c) *Error-correcting code process*, decoding the PUF response with error bits using helper data to generate a corrected response, which is then hashed (e.g., via SHA-256) to produce a stable cryptographic key.

To clarify and provide the readers a broad image on the robustness of digit bit generation, we have now made the following revision in the revised manuscript:

(Line 112) Here we note the error bits generated may be further corrected by hardware (e.g. using high precision ADC or alternatively other analogue-digital conversion approaches by FPGA and microprocessors) and algorithm designs (e.g. temporal majority voter, median value by multiple measurements, and error-correcting code; Fig. S9).¹³

5) PUFs are a promising hardware security solution. Looking forward to future practical applications, since the method as presented by the authors is potentially scalable, how can the method be further improved to promote its real-world implementation and industrialization process? The authors may want to elaborate on this for a larger impact of the work.

We appreciate the reviewer for the favorable comment on our work. Looking forward to future practical applications, we consider the following four major improvements to translate our lab-scaled carbon nanotube based PUFs:

- *Miniaturization*: Our PUFs are fabricated via a standard photolithographic patterning process. Limited by the lab-scaled research-use photolithographic system in our work, the feature size of the carbon nanotube transistors is $\sim 5\text{-}20\ \mu\text{m}$. Looking forward to realizing practical applications, the fabrication shall be miniaturized using industrial-scale advanced photolithographic patterning systems, e.g. DUV and EUV lithography. This can achieve PUFs with a higher density of transistors and low power consumption for real-world use.
- *CMOS integration*: On top of miniaturization using advanced photolithography, the PUF fabrication shall be compatible with CMOS manufacturing processes towards PUF integration with the analog front end and digital logic. It is expected to achieve carbon nanotube deposition and selective etching into back-end-of-line (BEOL) via standard CMOS manufacturing processes. Through challenging, successful demonstrations on counterpart electronics development prove the feasibility of CMOS integration, e.g. carbon nanotube microprocessors reported in Hills, Gage, et al. *Nature* 572.7771 (2019): 595-602.
- *System design*: At the system level towards practical PUF design and development, multi-channel, pipelined ADCs with on-chip buffering paired with lightweight ASIC accelerators for error-correction coding may be designed and integrated to define and develop operational PUF hardware systems. In addition, as discussed, for further robust and high-frequency PUF operations, FPGA and microprocessors may be adapted and integrated to develop the operational PUF hardware systems.
- *Algorithm design*: Finally, towards practical PUF applications, algorithms shall be co-designed to operate the PUFs. This can involve PUF operation instruction design from reconfiguration, challenge and response, and PUF generation to the end applications. Besides, as discussed, real-time calibration and adaptive threshold-tuning routines may

be designed to ensure a robust operation of the PUFs.

To provide the readers a further image of our PUFs towards their future practical applications, we have now made the following discussion in the revised manuscript:

(Line 280) Looking forward to bridging the lab-to-fab gap for our PUFs towards the real-world applications, our PUFs expect to adapt to industrial-scale advanced photolithographic systems for the manufacture of PUFs with high density of miniaturized transistors and low power consumption. The manufacture also expects to meet CMOS integration towards integral PUFs with the analog front end and digital logic fabricated and integrated onto one single chip. At the system level, multi-channel pipelined ADCs with on-chip buffering, FPGA, and microprocessors may be integrated for operating the PUFs. Furthermore, hardware-algorithm co-designs in operation instructions and error correction strategies are demanded for practical operations of the PUFs.

Reviewer #2 (Remarks on code availability):

The code is enough and sufficient. I can see the results on Code Ocean and they are right to me.

Reviewer #3 (Remarks to the Author):

In this work, Liu et al. present the development of physical unclonable functions (PUFs) based on reconfigurable carbon nanotube (CNT) charge-trapping transistors. The fabrication and characterization of the transistors are thoroughly described, demonstrating significant potential for PUF development. After that, the authors evaluate the resilience of the developed PUFs against attacks. While they claim that the high entropy of the PUFs ensures the security of edge communications, this aspect of the work remains unclear.

The authors also include a simulation-based application within a Vehicular Ad-Hoc Network (VANET) using the OMNeT++ platform. Although this scenario is interesting, it does not clearly demonstrate how the proposed PUFs enhance robustness or privacy compared to existing solutions.

In summary, the generation of PUFs using reconfigurable CNT charge-trapping transistors is promising and shows strong potential for hardware-based security solutions. However, the practical application discussed in this work is somewhat ambiguous and should be clarified.

We thank the reviewer for the favorable comments on our work. Here we address the queries raised in our responses below point by point:

- **The process used to configure and reset individual transistors should be explained in more detail, with experimental results to ensure reproducibility.**

To operate our PUFs (Fig. 2), the first step is to reset and configure a PUF to a state with voltage pulses V_{gs} applied onto the gates of the 9 individual transistors of the PUF, the second step is to challenge the PUF with voltage pulses V_{in} applied onto the common drain of the PUF, and the third step is to convert the current outputs I_{out} read from the sources of these 9 transistors into 12-bit digit numbers.

Figure R4 schematically illustrates the resetting and configuration of the individual transistors. Briefly, positive pulses V_{gs} (9 V) are first applied onto the gates of all the individual transistors to reset all the transistors to the initial high conductance states, and then negative pulses V_{gs} of different amplitudes are applied onto the gates of all the individual transistors to configure the transistors to the states as programmed. This completes the resetting and configuration of the individual transistors, i.e. the resetting and configuration of the PUF.

Following the configuration, the PUF can be operated for challenge and response. Briefly, voltage pulses V_{in} (0.5 V, duration 10 μ s) are applied onto the common drain of the transistors as the challenge, and current outputs I_{out} are read from the source of the individual transistors as the responses.

Figure R4. Caron nanotube transistor operation of the PUFs. (a) Diagram and (b) the corresponding pulse profiles showing the operation of the transistors. Voltage pulses V_{gs} are first applied onto the gate of all the individual transistors for resetting and configuration, and voltage pulses V_{in} are then applied onto the common drain of the transistors as the challenge. Current outputs I_{out} are read from the source of the individual transistors as the responses.

We have conducted rigorous test to study the reproducibility for resetting and configuring of our transistors. Here we show the cycle-to-cycle pulsed modulation of the transistors (Fig. R5). In this test, for a typical transistor, alternating positive-negative voltage pulses V_{gs} are applied to reset and configure the transistor from cycle to cycle, and the conductance modulating behavior is probed. To specify, 30 consecutive positive pulses V_{gs} (amplitude 3 V, 10 μ s) are applied to reset the transistor, and 30 consecutive negative pulses V_{gs} (amplitude 3 V, 10 μ s) are applied to configure the transistor. As demonstrated, across the 19,000 consecutive alternating pulse modulations, the conductance variation is maintained at a low level (<6.8%). This low cycle-to-cycle variation under long-term consecutive stimulations proves the reproducibility for the resetting and configuration operation of our transistors and hence, the reproducibility in operation of our PUFs. Indeed, as we prove in above Fig. R1 and R2, our individual transistors and PUFs prove low error bit generations.

Figure R5. Cycle-to-cycle operational reproducibility of the carbon nanotube transistors.

(a) The conductance profile probed from a typical transistor when modulated with alternating positive-negative voltage pulses V_{gs} . 30 consecutive positive pulses V_{gs} (amplitude 3 V, duration 10 μs) are applied to reset the transistor, and 30 consecutive negative pulses V_{gs} (amplitude -3 V, duration 10 μs) are applied to configure the transistor. The red dots correspond to the conductance in configuring, and the blue dots correspond to the conductance in resetting. The cycle-to-cycle variation in conductance modulation is $<6.8\%$. A drain voltage V_{in} of 0.5 V DC is applied to probe the conductance. (b) Zoomed-in conductance profile.

We hope our above responses have addressed the reviewer queries towards the resetting and configuration operation of the transistors, and the operational reproducibility. To clarify, we have now included the following discussion Fig. R4 and Fig. R5 in the revised manuscript and supporting information:

(Line 86) In addition, we demonstrate that the transistors allow cycle-to-cycle stable resetting and reconfiguration operations (Fig. S3).

• The authors claim that the proposed PUFs outperform those in previous studies (Fig. S1). However, a detailed, quantitative comparison should be added in the main manuscript, including benchmarking metrics for the different key parameters.

In the initial submission, in Fig. S1, we show that our transistors can be operated in a wide

temperature window from a cryogenic temperature (e.g. 100 K) to a high temperature (e.g. 400 K). This outperforms previous reports on PUFs, as we compare in Table. S1. In Table. S1, we compare our PUFs with state-of-the-art PUFs in terms of the reconfigurability, mechanism, physical unclonability metrics, work temperature, power consumption, cracking time, and attacking resilience.

As pointed out by the reviewer, to help the readers conveniently compare our PUFs with the state-of-the-art reports, we have included the following discussion on quantitative comparison in the revised manuscript. We have also included below Fig. R6 in the revised manuscript to help visualize the comparison. In Fig. R6, we quantitatively compare some of the key metrics, including reconfigurability, physical unclonability (using correlation coefficients (CC) as the example), cracking time, AI attacking resistance, and work temperature.

Figure R6. Comparison of our PUFs with state-of-the-art reports. Radar plot of our PUFs with quantitative comparison with memristor PUFs, silicon PUFs, and 2D materials PUFs in the correlation coefficients (CC), reconfigurability, cracking time, AI attacking resistance, and work temperature.

To clarify, we have now made the following revision in the revised manuscript:

(Line 215) As discussed above, our PUFs prove performance outperforming previous reports (e.g. ref 9,17,26) in, for instance, reconfigurability, physical unclonability, and attacking resilience (Table S1). We illustrate quantitative comparison in some of the metrics in Fig. 5h, g. This well demonstrates the capability of our PUFs in securing hardware security in edge scenarios.

- The sentence “PUF all prove a stable operation after even 120 days in ambient conditions” is unclear. I assume the authors mean that the devices were tested up to 120 days and demonstrated stable operation throughout that period, although the stability

may extend beyond that. I recommend rephrasing the sentence to clarify this point.

We appreciate the reviewer for pointing this out. Yes, indeed, by this sentence we aimed to express that our transistors were tested up to 120 days and demonstrated stable operation throughout that period, and that the stability might extend beyond that period.

To clarify, we have now made the following revision in the revised manuscript:

(Line 116) ...the PUFs prove a long lifetime, as demonstrated in Fig. S10 where we show the transistors of a typical PUF are tested up to 120 days and demonstrate stable operation throughout the period. The operational stability may, however, extend beyond this.

• Line 120: Sentence “the primitives all present an n-HD of ~0.5”, is imprecise. Please provided the obtained values. Same for entropy and CC values.

We appreciate the reviewer for pointing this out as well. We provide the detailed values for the entropy, n-HD and CC in below Table R1. To clarify, we have now also included Table R1 in the revised supporting information.

Table R1. Obtained values of our reconfigurable PUFs.

	Entropy	n-HD	CC
Fig. 3a	0.98 ± 0.03	0.49 ± 0.05	0.02 ± 0.10
Fig. 3b	0.98 ± 0.03	0.48 ± 0.06	0.03 ± 0.11
Fig. 3c	/	0.48 ± 0.09	/
Fig. 3d	Index #1:	Index #1:	Index #1:
	0.98 ± 0.03	0.48 ± 0.06	0.03 ± 0.11
	Index#2:	Index#2:	Index#2:
	0.95 ± 0.04	0.46 ± 0.05	0.06 ± 0.11
	Index#3:	Index#3:	Index#3:
	0.93 ± 0.05	0.42 ± 0.05	0.11 ± 0.10
	Index#4:	Index#4:	Index#4:
	0.90 ± 0.07	0.39 ± 0.04	0.17 ± 0.09
	Index#5:	Index#5:	Index#5:
	0.86 ± 0.07	0.36 ± 0.04	0.18 ± 0.09
	Index#6:	Index#6:	Index#6:
	0.81 ± 0.08	0.32 ± 0.03	0.23 ± 0.07

• It would be interesting to also include in Table S1 some PUF based in memristors, as [1].

We thank the reviewer for the suggestion. Memristors with intrinsic stochasticity are promising device candidates for developing PUFs. We have now included the suggested reference along with some other state-of-the-art reports on memristor based PUFs in Table S1 in the revised supporting information.

• The application in the vehicular network should be more thoroughly introduced, with a clearer explanation of how the proposed technology improves upon the current state-of-the-art, including quantifiable metrics.

In our work, having proved the capability of our PUFs to enhance hardware security, we explore the potential application in self-driving. This exploration is conducted on OMNeT++ platform. OMNeT++ is a widely adopted open-source platform for building network simulators to implement and study real-time road traffic and vehicle communication (Varga, A. OMNeT++. in *Modeling and Tools for Network Simulation*, Springer Berlin Heidelberg, Berlin, Heidelberg, 2010).

As schematically illustrated in Fig. R7, when building the vehicular network on OMNeT++, each vehicle is equipped with a carbon nanotube PUF chip, and the secured communication among the vehicles can be phased into *Reconfiguration*, *Authentication*, and *Communication* stages.

- *Reconfiguration* (Fig. R7a): A vehicle equipped with a PUF chip initiates a request to the trust authority to update its identity. The trust authority checks and verifies the current identity of the vehicle and sends a challenge to the vehicle. Upon receiving the challenge, the PUF equipped on the vehicle generates a new challenge-response pair to create a fresh, unclonable identity which is then verified and stored by the trust authority. This *Reconfiguration* process is designed to ensure safe and dynamic identify update of the vehicles in the vehicular network. The *Reconfiguration* implementation protocol is detailed in Fig. S27.
- *Authentication* (Fig. R7b): The trust authority issues a challenge to the PUF equipped on the vehicle, and the PUF upon receiving the challenge generates and returns its response to the trust authority. As such, the vehicle allows the trust authority to verify its identity by checking the response sent back. This completes a mutual-verification handshake in less than 1 ms, an order of magnitude faster than the conventional approaches (Wang, Shaoqiang, et al. *Electronics* 13.8 (2024): 1418). Due to the unique, unclonable nature of the PUFs, attacking becomes virtually impossible, enabling secure access control. The *Authentication* implementation protocol is detailed in Fig. S28.

- *Communication* (Fig. R7c): Using the communication between Vehicle A and B as an example, Vehicle A sends message with its PUF-derived session token and a timestamp to Vehicle B, and Vehicle B then forwards the packet to the trust authority for a sub-50- μ s timestamp verification. Upon receiving the verification, Vehicle B generates and sends message with its PUF-derived session token and a timestamp to Vehicle A. Again, Vehicle A then forwards the packet to the trust authority for verification. Upon receiving the verification, Vehicle A completes a mutual-verification handshake with Vehicle B before their communication. As such, vehicles use PUFs to establish secure communication with timestamps and secure cryptographic tokens verified by the trust authority. This ensures the hardware integrity and prevents replay attacks. The *Communication* implementation protocol is detailed in Fig. S29.

In this modelling, PUFs are embedded in the vehicular network by embedding a *PUF Module* in each vehicle module. Specifically, the *PUF module* is allowed to feature the characteristics of our carbon nanotube PUFs in generating 108-bit responses upon challenges, including the reconfigurability and input-output response times.

To specify, during the modelling, *Simulation of Urban MObility* (SUMO) simulates the movement of a self-driving vehicle. When the vehicle enters a set range, wireless vehicle communication can be activated, and the *PUF module* begins to work. For *Communication*, the trust authority sends a challenge to the vehicle, and the *PUF module* in the vehicle outputs a 108-bit response. Since the *PUF module* simulates the response behavior of our carbon nanotube PUFs to the challenge, there is a time delay of about 1 ms in generating the 108-bit response. Time delay is a key parameter in the *Communication* process, which can affect the freshness of the message and the success rate of attacking. According to the above discussion and results we show in Fig. 6, the median end-to-end time delay measured by the time-accurate event-driven simulator is about 12 ms (~100 ms in the worst case for up to 100 vehicles). Attacking the response in such a short time is nearly impossible (Fig. 5e), so it can ensure the security of message transmission. In addition, the 108-bit response is embedded in the message transportation in the *Vehicular ad hoc network* (VANET) after passing through the Elliptic Curve Cryptography (ECC) based encryption. Based on the *Communication* implementation protocol (Fig. S29), the ECC-based encryption cost (around 0.87 ms per vehicle) is very low, so it can enable lightweight communication.

The complete secured communication between the vehicles, including the *Reconfiguration*, *Authentication*, and *Communication* stages, are modelled and presented in Movie S1. The result demonstrates that our PUFs can be embedded into the existing protocol flows and deliver

hardware security with substantially reduced communication and computational overhead as well as communication delays (Fig. 6). For example, we approach achieves a typical time delay of ~ 12 ms for networks of 10 to 100 vehicles, with a maximum delay of ~ 100 ms, well within the 50 ms self-driving requirement, and enables lightweight authentication and communication with only 324 bits per vehicle for communication and 1,544 bits for authentication, alongside a low computational cost of ~ 0.87 ms per vehicle.

Figure R7. Flowchart of how PUFs play an important role in the vehicular network systems. The whole reconfiguration, authentication, communication process where PUF chips in the car account in the vehicular network systems are demonstrated.

To clarify and clearly illustrate the building of the vehicular network on OMNeT++ with our PUFs to the readers, we have now included the following revisions and Supplementary Note 1,

as well as Fig. R7, in the revised manuscript and supplementary information.

(Line 226) Here we explore self-driving vehicular network modelling with PUF-based key exchange protocols embedded to secure vehicle communication on OMNeT++ platform (Fig. 6a). OMNeT++ is a widely adopted general platform for building network simulators to implement and study real-time road traffic and vehicle communications.²⁹ In this modelling, the map of Central Hong Kong is used, and the secured communication among the vehicles is in general phased into *Reconfiguration*, *Authentication*, and *Communication* stages (Fig. S27). Briefly, *Reconfiguration* is designed to allow the vehicles to update the identities, *Authentication* is to allow the trust authority to verify the identities of the vehicles, and *Communication* is to allow the vehicles to authenticate their identities before establishing secure communication. See Supplementary Note 1 and Fig. S28-30 for the detailed implementation processes and protocols, where PUF-based keys are exchanged among the vehicles and the trust authority for the mutual-verification and authentication handshakes. Particularly, in this modelling, PUF is embedded a *PUF Module* in each vehicle module to 108-bit responses upon challenges. This *PUF Module* is allowed to feature the characteristics of our carbon nanotube PUFs including the reconfigurability and input-output response times. As such, a time delay of ~1 ms is cost by the *PUF Module* in generating the 108-bit responses. Note that a lightweight public-key cryptosystem Elliptic Curve Cryptography (ECC)²⁹ is used here for PUF-based key exchange at the edge scenario.

(Line 261) In summary, our PUF-based key exchange protocol achieves a typical time delay of ~12 ms for the networks and enables lightweight authentication and communication with only 324 bits per vehicle for communication and 1,544 bits for authentication, alongside a low computational cost of ~0.87 ms per vehicle.

• In line 212, the authors affirm that they have adapted “a PUF-based key exchange protocol, following authentication and key generation in Fig. 5”. As the PUF is being used to generate cryptographic keys, it would be convenient to study if the proposed PUFs primitives pass the NIST SP800-22 tests [2].

We thank the reviewer for pointing this out. We have now used NIST SP800-22 to test the randomness of the cryptographic keys generated by our PUFs. According to the test (Table R2), the cryptographic keys generated can successfully pass all the randomness tests. This confirms the viability of using our PUFs for cryptographic applications.

Table R2. NIST SP800-22 test results of keys based on our PUFs.

	P-value	Proportion	Success	Post-processing
Approximate entropy	0.299	90/90	Success	No
Block frequency	0.740	90/90	Success	No
Cumulative sums	0.987	90/90	Success	No
FFT	0.226	88/90	Success	No
Frequency	0.844	90/90	Success	No
Linear complexity	0.388	89/90	Success	No
Longest run	0.557	90/90	Success	No
Non overlapping template	0.911	89/90	Success	No
Overlapping template	0.168	88/90	Success	No
Random excursions	0.503	90/90	Success	No
Random excursions variant	0.314	90/90	Success	No
Rank	0.717	90/90	Success	No
Runs	0.168	89/90	Success	No
Serial	0.557	89/90	Success	No
Universal	0.853	90/90	Success	No

To clarify and provide the readers detailed information on the randomness of the cryptographic keys generated by our PUFs, we have now made the following discussion in the revised manuscript, and included Table R2 in the revised supporting information:

(Line 199) Note that the key generated can pass all the NIST SP800-22 tests, proving their high-level randomness (Table S3).

- Regarding the cryptosystems used in the example, Figure S21 presents RSA as the public-key cryptosystem used in the example, while in Figures S22 to S24, Elliptic Curve Cryptography (ECC) is used for the implementation of the protocols. This should be clarified. In addition, a reference to the article/ articles where these authentication protocols are presented is required.

In Fig. S21 (now Fig. S25), we illustrate a PUF-seeded Rivest Shamir Adleman (RSA) scheme for securing data traffic between an edge device and a back-end server. RSA is a commonly

used, general data traffic and encryption method for establishing secure channels (*Menezes, A. J., van Oorschot, P. C., & Vanstone, S. A. (1996). Handbook of Applied Cryptography. CRC Press*). Particularly, the simplicity and compatibility with existing infrastructures of RSA make it well suited for end-use devices with insufficient computational resources, providing well-understood, standards-based means of confidentiality and authentication without requiring proprietary hardware. In this consideration, we use RSA for securing the data traffic in our work. Here the 108-bit response generated by our PUF can be converted via a fuzzy extractor into a symmetric session key or identifier, which can be then encrypted or digitally signed using RSA.

In Fig. S22-24 (now Fig. S27-29), we illustrate PUF-based lightweight mutual-authentication and key-agreement protocols built on Elliptic Curve Cryptography (ECC). ECC is a public-key cryptography method that uses the mathematical properties of elliptic curves to provide strong security with smaller key sizes (*Delvaux et al., IEEE Transactions on Information Forensics and Security*). This efficiency stems from the complex algebraic structure of elliptic curves that makes solving the underlying mathematical problems computationally difficult for attackers while allows faster computations and lower resource demands on devices. As such, ECC enables efficient mutual authentication and key establishment, making it ideal for resource-constrained edge devices in vehicular networks. In this consideration, we use ECC for securing the data traffic in our vehicular network modeling. Here, the 108-bit responses generated by our PUFs can be converted via a fuzzy extractor into a symmetric session key or identifier that can then be used in ECC-based protocols for mutual authentication and secure key agreement.

To clarify and address the possible queries from the readers towards the crypto strategies in the difference hardware security applications, we have now made the following revisions in the revised manuscript and supplementary information.

(Line 202) See Fig. S25 for the secure data communication protocols using PUF-based private and public keys **where a public-key cryptosystem Rivest Shamir Adleman (RSA) is used for the data traffic encryption.**¹⁷

(Line 240) **Note that a lightweight public-key cryptosystem Elliptic Curve Cryptography (ECC)**²⁹ **is used here for PUF-based key exchange at the edge scenario.**

• The discussion of how PUFs integrate into common system architectures (such as in Fig. S21) should be expanded, indicating the implications and potential limitations.

Integrating our lab-scaled carbon nanotube based PUFs into the common system architectures (e.g. data traffic security between end-users and back-end server as presented in Fig. S21(now

Fig. S25) in real-world needs to bridge the lab-to-fab gap. In the real-world applications, our PUFs can significantly reduce the attacking risks from physical tampering and cloning, enabled by their hardware-intrinsic unclonable nature. Furthermore, the dynamic reconfiguration and PUF primitive regeneration capability of our PUFs can support one-time encryption and secure key exchange protocols, making the system highly resistant to side-channel and modeling attacks. We consider the following four major potential limitations to address in translating our PUF technology:

- *Miniaturization*: Our PUFs are fabricated via a standard photolithographic patterning process. Limited by the lab-scaled research-use photolithographic system in our work, the feature size of the carbon nanotube transistors is $\sim 5\text{-}20\ \mu\text{m}$. Looking forward to realizing practical applications, the fabrication shall be miniaturized using industrial-scale advanced photolithographic patterning systems, e.g. DUV and EUV lithography. This can achieve PUFs with a higher density of transistors and low power consumption for real-world use.
- *CMOS integration*: On top of miniaturization using advanced photolithography, the PUF fabrication shall be compatible with CMOS manufacturing processes towards PUF integration with the analog front end and digital logic. It is expected to achieve carbon nanotube deposition and selective etching into back-end-of-line (BEOL) via standard CMOS manufacturing processes. Through challenging, successful demonstrations on counterpart electronics development prove the feasibility of CMOS integration, e.g. carbon nanotube microprocessors reported in Hills, Gage, et al. *Nature* 572.7771 (2019): 595-602.
- *System design*: At the system level towards practical PUF design and development, multi-channel, pipelined ADCs with on-chip buffering paired with lightweight ASIC accelerators for error-correction coding may be designed and integrated to define and develop operational PUF hardware systems. In addition, as discussed, for further robust and high-frequency PUF operations, FPGA and microprocessors may be adapted and integrated to develop the operational PUF hardware systems.
- *Algorithm design*: Finally, towards practical PUF applications, algorithms shall be co-designed to operate the PUFs. This can involve PUF operation instruction design from reconfiguration, challenge and response, and PUF generation to the end applications. Besides, as discussed, real-time calibration and adaptive threshold-tuning routines may be designed to ensure a robust operation of the PUFs.

To present the readers our outlook on integrating our lab-scaled PUFs into the common system architectures in real-world applications, we have now made the following discussion in the revised manuscript:

(Line 280) Looking forward to bridging the lab-to-fab gap for our PUFs towards the real-world applications, our PUFs expect to adapt to industrial-scale advanced photolithographic systems for the manufacture of PUFs with high density of miniaturized transistors and low power consumption. The manufacture also expects to meet CMOS integration towards integral PUFs with the analog front end and digital logic fabricated and integrated onto one single chip. At the system level, multi-channel pipelined ADCs with on-chip buffering, FPGA, and microprocessors may be integrated for operating the PUFs. Furthermore, hardware-algorithm co-designs in operation instructions and error correction strategies are demanded for practical operations of the PUFs.

- The term “hash function” is misspelled in several places (e.g., page 5, line 187).

We appreciate the reviewer for pointing this out. We have corrected this mistake in the revised manuscript and supplementary information.

References

- [1] Ibrahim, Hebatallah M., Heorhii Skovorodnikov, and Hoda Alkhzaimi. "Resilience evaluation of memristor based PUF against machine learning attacks." *Scientific reports* 14.1 (2024): 23962.
- [2] Pareschi, Fabio, Riccardo Rovatti, and Gianluca Setti. "On statistical tests for randomness included in the NIST SP800-22 test suite and based on the binomial distribution." *IEEE Transactions on Information Forensics and Security* 7.2 (2012): 491-505.

Reviewer #3 (Remarks on code availability):

I reviewed the code but did not run it, as authentication was required on the platform where the authors uploaded the code, which could compromise the reviewers' anonymity. In any case, the information provided appears to allow the application to be installed and run on a self-managed computer.

For the convenience of the reviewer, we have now uploaded our code on Github as well:

<https://github.com/ghhucuhk/Chip-scale-reconfigurable-carbon-nanotube-physical-unclonable-functions>.

The reviewer may want to refer to the README file for the detailed introduction on the code and environmental configuration.

Reviewer #4 (Remarks to the Author):

We thank the reviewer for the evaluation of our work and providing the detailed, constructive comments. Please refer to our above point-by-point responses.